# In Vitro Study of Two Edible Polygonoideae Plants: Phenolic Profile, Cytotoxicity, and Modulation of Keap1-Nrf2 Gene Expression

**DOI:** 10.3390/foods10040811

**Published:** 2021-04-09

**Authors:** Marina Jovanović, Dina Tenji, Biljana Nikolić, Tatjana Srdić-Rajić, Emilija Svirčev, Dragana Mitić-Ćulafić

**Affiliations:** 1Institute of General and Physical Chemistry, Studentski trg 12-14/V, 11000 Belgrade, Serbia; 2Faculty of Sciences, University of Novi Sad, Trg Dositeja Obradovića 3, 21000 Novi Sad, Serbia; dinatenji@gmail.com (D.T.); emilija.svircev@dh.uns.ac.rs (E.S.); 3Faculty of Biology, University of Belgrade, Studentski trg 16, 11000 Belgrade, Serbia; biljanan@bio.bg.ac.rs (B.N.); mdragana@bio.bg.ac.rs (D.M.-Ć.); 4Institute of Oncology and Radiology of Serbia, Pasterova 14, 11000 Belgrade, Serbia; tsrdic@gmail.com

**Keywords:** edible plants, Polygonoideae, phenolic profile, doxorubicin, apoptosis, cell cycle, Keap1-Nrf2 expression

## Abstract

*Polygonum aviculare* and *Persicaria amphibia* (subfam. Polygonoideae) are used in traditional cuisines and folk medicine in various cultures. Previous studies indicated that phytochemicals obtained from Polygonoideae plants could sensitize chemoresistant cancer cells and enhance the efficacy of some cytostatics. Here, the cytotoxic properties of chemically characterized ethanol extracts obtained from *P. aviculare* and *P. amphibia*, individually and in combination with doxorubicin (D), were determined against hepatocarcinoma HepG2 cells. Phenolic composition, cell viability, cell cycle, apoptosis, and the expression of Keap1 and Nrf2 were examined by following methods: LC-MS/MS, LC-DAD-MS, MTT, flow cytometry, and qRT-PCR. Extracts were rich in dietary polyphenolics. Synergistic cytotoxicity was detected for extracts combined with D. The observed synergisms are linked to the interference with apoptosis, cell cycle, and expression of Keap1-Nrf2 genes involved in cytoprotection. The combined approach of extracts and D could emerge as a potential pathway of chemotherapy improvement.

## 1. Introduction

Widespread throughout Europe, Asia and the Americas, wild plants *Polygonum aviculare* and *Persicaria amphibia* (syn. *Polygonum amphibium*), subfamily Polygonoideae, are used in traditional cuisines and folk medicine in various cultures [1,2]. *P. aviculare*, known as the common knotweed, is edible and used as a Korean salad plant, an Australian honey plant, and a traditional Vietnam culinary herb [3,4,5]. In the USA, *P. amphibia,* popularly known as water smart weed, has been utilized in soft drink preparation [2]. Described as healing weeds, these plants are widely used as a home remedy to treat ailments such as stomach pains and diarrhea [1,2,6,7]. Concerning Serbia, *P. aviculare* is mainly used as an appetite stimulant [8]. Importantly, in the folk medicine of China and Austria *P. aviculare* and *P. amphibia* are employed to treat some types of cancer [9,10]. Chemical properties of these plants have been thoroughly investigated in recent decades and acquired data showed that their extracts are rich in flavonoids, sesquiterpenoids, and tannins, thus, justifying their ethnopharmacological use and contributing to their recognition in contemporary pharmacology [1,2,11,12]. Although a limited number of pharmacological studies regarding these herbs are available, some of them indicate that these plants and their active compounds could be used for the treatment of various diseases in clinical medicine, including diabetes and some types of cancer [13,14,15]. Furthermore, epidemiological evidence has demonstrated that a diet rich in natural bioactive compounds could decrease the risk of cancer development and could be used in chemoprevention. The discovery of plant-derived drugs has emerged as a potential pathway in the search for chemotherapeutics owing to the accepted assumption that plant medicaments are safer than their synthetic counterparts. In addition, toxic and other unfavorable effects of synthetic anticancer drugs have been widely noted [2,16,17]. Chemotherapy treatment with anthracycline drugs, such as doxorubicin (D), results in high hepatotoxicity. Apart from the numerous side effects, the medical application of D is also limited due to the frequent development of resistance in tumor cells [18,19]. Considering that D could rely on an increase in the free radical production to exhibit its effect, the reduction of antioxidant defense could initially make the cancerous cells susceptible to chemotherapeutics [20,21]. Importantly, numerous cancerous cells possess increased endogenous antioxidant defense due to the constitutive overexpression of the nuclear factor erythroid 2-related factor 2 (Nrf2) related to the disruption of Kelch-like ECH-associated protein 1 (Keap1) [19,22]. Keap1 acts as a negative regulator of Nrf2, and hence, it may act as a tumor suppressor in cancer cells. Nrf2 is the redox-sensitive transcription activator that regulates the expression of a large number of cytoprotective enzymes [23]. Thereby, Nrf2 has been proposed as a novel therapeutic target to overcome chemoresistance in various types of cancer, including hepatocellular carcinoma (HCC) [22]. Moreover, it has been observed that some phytochemicals have the potential to sensitize chemoresistant HCC through the suppression of Nrf2 [22].

Therefore, in this work, the phenolic profiles and cytotoxic properties of ethanol extracts of aerial parts of *P. aviculare* (POA) and *P. amphibia* (PEA) were explored, as well as their potential to modulate the response of human hepatocellular carcinoma cells (HepG2) to D, the most widely used cytostatic in HCC treatment [16]. The cytotoxic properties of extracts, alone and combined with D, were estimated by MTT assay and flow cytometric analysis. The potential of co-treatments of extracts and D to influence Nrf2 and Keap1 expression was assessed by qRT-PCR. Thus, this study, in a comprehensive manner, investigated cytotoxic properties of POA and PEA. Using both plant extracts and cytostatic D, it aimed to present the benefits of the combined approach in order to make an initial step in chemotherapy improvement. 

## 2. Materials and Methods

### 2.1. Materials

Reference standards of the secondary metabolites used in LC-MS/MS analysis were obtained from Sigma–Aldrich Chem (Steinheim, Germany): 4-hydroxy-benzoic acid, 2,5-dihydroxybenzoic acid, vanillic acid, gallic acid, cinnamic acid, caffeic acid, trans-ferulic acid, 3,4-dimethoxycinnamic acid, D-(−)-quinic acid, umbelliferon, matairesinol, secoisolariciresinol, chlorogenic acid, predominantly trans, quercetin dihydrat, (+)-catechin hydrate, baicalein, genistein, daidzein, baicalin, syringic acid, p-coumaric acid (predominantly trans isomer), 2-hydroxycinnamic acid (predominantly trans), sinapic acid (predominantly trans isomer), scopoletin, (−)-epicatechin, quercetin-3-*O*-beta-D-glucoside, quercitrin-hydrate, (−)-epigallocatechin gallate; Roth/Carl Roth GmbH/Rotichrom^®^: protocatechuic acid, esculetin, apigenin, apigenin-7-*O*-glucoside, hyperoside, chrysoeriol, amentoflavone trihydrate, apiin; Chromadex (Santa Ana, CA, USA): kaempherol, kaempferol 3-*O*-glucoside, naringenin, isorhamnetin; Extrasynthese Genay Cedex France: luteolin, luteolin-7-*O*-glucoside, and from Fluka Chemie GmbH (Buchs, Switzerland): myricetin, vitexin, rutin, trihydrate. HPLC gradient-grade methanol was purchased from J. T. Baker (Deventer, The Netherlands) and p.a. formic acid from Merck (Darmstadt, Germany). Folin and Ciocalteu’s Phenol Reagent (FC) was provided by Sigma–Aldrich, while sodium carbonate and aluminium(III) chloride were purchased from Centrohem (Stara Pazova, Serbia) and Kemika (Zagreb, Croatia), respectively. William’s medium, fetal bovine serum (FBS), penicillin-streptomycin mixture, phosphate-buffered saline (PBS), trypsin from porcine pancreas, dimethyl sulfoxide (DMSO), and 3-(4,5-dimethylthiazol-2-yl)-2,5-diphenyltetrazolium bromide (MTT) were purchased from Sigma-Aldrich (Steinheim, Germany). Reagents for apoptosis and cell cycle assay were obtained from Invitrogen Life Technologies^TM^ (FITC-AnexinV, Binding puffers 2x, Rnase A Pure Link^TM^, Waltham, MA, USA) and 7- amino actinomycin D was provided from Pharmingen^TM^ (Franklin Lakes, NJ, USA). Trizol reagent, Power SYBR green PCR master mix and specific primers for qRT-PCR were obtained from Invitrogen Life Technologies^TM^ (Carlsbad, CA, USA). Doxorubicin (D, Cas. No. 25316-40-9) was provided by Actavis, S.C. Sindan-Pharma S.R.L. (Buchurești, Romania). All the other chemicals and reagents were purchased from local companies and were of a molecular biology grade. 

### 2.2. Plant Material, Extracts Preparation, and Chemical Analysis

Aerial parts of *P. aviculare* and *P. amphibia* were collected at Vlasina Lake (N42°42′40.09″ E22°20′32.942″) in Serbia. Plant materials were identified and the voucher specimens were deposited at the Herbarium of Department of Biology and Ecology, Faculty of Natural Sciences, University of Novi Sad, Serbia (BUNS Herbarium; voucher numbers for *P. aviculare* and *P. amphibia* are 2-1669 and 2-1691, respectively).

Extracts were prepared by the maceration of air dried and powdered aboveground plant material (10 g) with 80% ethanol (100 mL) for 72 h under constant stirring at room temperature. Extracts were removed from plant material by filtration and after vacuum drying, yield of dry raw extracts were: 1.31 g/13.1% (*P. aviculare*) and 1.32 g/13.2% (*P. amphibia*). Raw extracts were suspended in water and purified by liquid–liquid extraction with petroleum ether, to remove chlorophyll and other ballasts. Defatting of the extracts with petroleum ether can lead to losses of the compounds of interest [24,25], so the petroleum ether layer was washed with methanol and methanol fraction pooled with water layer. After purification, herb extracts yield decreased for 1.1% (1.20 g; *P*. *aviculare*) and for 0.8% (1.23 g; *P*. *amphibia*). Vacuum dried (<45 °C) purified extracts were dissolved in DMSO to a final concentration of 100 mg/mL.

The phytochemical profiles of *P. aviculare* and *P. amphibia* were evaluated by measuring the total phenolic content (by means of Folin–Ciocalteu reagent under alkaline conditions) and total flavonoids content (based on aluminum–flavonoids complex formation). Detailed procedure of these two spectrophotometric methods was previously published by Beara et al. [26]. Quantitative LC-MS/MS analysis of selected 45 secondary metabolites was carried out according to the previously reported method [27]. Standard mixture (containing 45 phenolics) was double diluted with mobile phase solvents: A (0.05% aqueous formic acid): B (methanol), in 1:1 ratio, to obtain fifteen working standards (from 25,000 ng/mL to 1.53 ng/mL). Extracts were diluted also with solvents A:B (1:1) to a final concentration of 2 mg/mL. Samples and standards were analyzed using Agilent Technologies 1200 Series high-performance liquid chromatograph coupled with Agilent Technologies 6410A Triple Quad tandem mass spectrometer with electrospray ion source (ESI), and controlled by Agilent Technologies MassHunter Workstation software—Data Acquisition (ver. B.03.01). The detailed procedure and method validation were published previously [27] (Appendix A).

Qualitative LC-DAD-MS analysis of extracts was performed on Agilent Technologies 1200 Series HPLC with DAD, coupled with Agilent Technologies 6410A Triple Quad tandem mass spectrometer with electrospray ion source, and controlled by Agilent Technologies MassHunter Workstation software—Data Acquisition (ver. B.03.01). Working solution of standard mixture-45 (1.56 μg/mL) and 5 μL of extracts (20 mg/mL diluted with A:B (1:1)) were injected into the system, with Zorbax Eclipse XDB-C18 (50 mm, 4.6 mm, 1.8 μm) rapid resolution column held at 50 °C. Mobile phase A (0.05% aqueous formic acid) and B (methanol) was delivered at flow rate of 0.8 mL/min in a gradient mode (0 min 20% B, 6.67 min 60% B, 8.33 min 100% B, 12.5 min 100% B, re-equilibration time 4 min). Eluted components were firstly recorded on diode array detector (DAD), full spectra in 190–700 nm range, chromatograms were acquired at 254 nm, 340 nm and 430 nm; and secondly on triple quadrupole mass spectrometer, using MS2Scan run mode (both, positive and negative ionization, m/z range of 120–1000 and fragmentor voltage of 80 V). ESI ion source parameters were as follows: nebulization gas (N_2_) pressure 40 psi, drying gas (N_2_) flow 9 L/min and temperature 350 °C, capillary voltage 4 kV.

### 2.3. Human Cell Line

The human cell line used in this study was hepatocellular carcinoma HepG2 (ATCC HB-8065, Manassas, VA, USA). HepG2 cells were grown in William’s medium, with 15% fetal bovine serum, 1% penicillin/streptomycin, and 2 mM of L-glutamine. The cell line was maintained in an incubator at 37 °C with 5.0% CO_2_ in a humidified atmosphere. The cells were sub-cultured at 90% confluence, twice a week, using 0.1% trypsin. Cell viability was determined by the trypan blue dye exclusion method. Cells in the logarithmic growth phase were used in all experiments.

### 2.4. Cytotoxicity and Drug Synergism Analysis

The cytotoxic effects of plant extracts and D, both as single compounds and in a mixture, were assessed by MTT assay, as described by Jovanović et al. [28]. HepG2 cells were seeded into 96-well plates at a density 2 × 10^4^ cells/well and incubated overnight with 5% CO_2_ at 37 °C. Further on, the cells were exposed to a series of two-fold dilutions of extracts and D in the ranges 4000–125 µg/mL and 22.8–0.712 µg/mL, respectively. To prepare mixtures of extracts and D, the highest concentrations of each substance were combined and subsequently diluted two-fold. This process was repeated until reaching 125 µg/mL and 0.712 µg/mL of extracts and D, respectively. After the incubation for 24 h, the medium with test substances was replaced with MTT (final concentration 0.5 mg/mL) and incubated for additional 3 h. At the end of incubation with MTT, the medium was removed, and the formazan crystals were dissolved in DMSO. The optical density was measured at 570 nm, using a micro-plate reading spectrophotometer (Multiskan FC, Thermo Scientific, Shanghai, China). Three independent experiments were conducted.

To evaluate the nature of interaction between extracts and D, combination index (CI) analysis was used, providing quantitative definition for the additive effect (CI = 1), synergism (CI < 1), and antagonism (CI > 1) in drug combinations [29]. The CI was calculated for IC_25_ and IC_50_ values of the mixtures, using the formula: CI = C_A_/IC_A_ + C_B_/IC_B_, where C_A_ is the concentration of the first test substance in the binary mixture; IC_A_ is the concentration of the first test substance alone; C_B_ is the concentration of the second test substance in the binary mixture; and IC_B_ is the concentration of the second test substance alone.

### 2.5. Flow Cytometry Analysis of Apoptosis and Cell Cycle Phase Distribution

Apoptotic cell death and analysis of the cell cycle phase distribution were analyzed using a fluorescence-activated cell sorting flow cytometer (FACS) (Calibur Becton Dickinson, Heidelberg, Germany) and Cell Quest computer software, according to manufacturer’s protocol. HepG2 (1 × 10^6^ cells/well) was cultured with plant extracts with and without D. Concentrations of tested substances were selected in accordance with the results of the MTT assay. IC_50_ values of extracts and D, individually and combined, were tested. Apoptotic or necrotic cell death was assessed after the 24 h treatment. As described by Srdic-Rajic et al. [30], cells were harvested, washed with PBS, and stained with Annexin V FITC and 7- amino actinomycin D (7-AAD). In brief, Annexin V FITC binds to the exposed phosphatidylserine of the early apoptotic cells, whereas 7-AAD labels the late apoptotic/necrotic cells, containing damaged membrane. The numbers of viable (annexin V FITC ^−^ 7AAD^−^), early apoptotic (annexin V FITC^+^ 7AAD^−^), and late apoptotic/necrotic (annexin V FITC^+^ 7AAD^+^) cells were determined.

The quantitative analysis of the proportion of cells in different cell cycle phases was performed after the treatment and incubation for 24 h. Cells were harvested and fixed with ice-cold 70% ethanol at −20 °C for 30 min. Subsequently, cells were resuspended in PBS containing propidium iodide and RNase A and incubated for 30 min at room temperature. The distribution of the cells was measured by FACS analysis, as previously described.

### 2.6. Real-Time Quantitative PCR (qRT-PCR) Analysis

In order to detect the expression pattern of Keap1 and Nrf2 genes in HepG2 cells, qRT-PCR analysis was conducted as described in Kaisarevic et al. [31], with minor modifications. For the experiment, the cells were seeded into 12-well plate (10^6^ cells/well) and, after 24 h, exposed in duplicates to the selected concentrations of combined extracts and D. The selected concentrations for this assay were the ones that induced 25% inhibitions of cell survival (IC_25_), considering that the test procedure requires high cell viability. After the 24 h treatment, the medium was removed, the cells were washed by PBS, and total RNA was extracted using trizol reagent according to supplier’s instructions. The quality and quantity of RNA was determined spectrophotometrically by BioSpec-nano (Schimadzu Corporation, Kyoto, Japan). Reverse transcription of each total RNA sample (2 μg) to cDNA was conducted using High-Capacity cDNA Reverse Transcription Kit with RNase inhibitor (Applied Biosystems). The reverse transcription reaction was conducted in the Veriti Thermal Cycler (Applied Biosystems), under the following incubation conditions: 10 min at 25 °C, 120 min at 37 °C, and 5 min at 85 °C. The expression level of Keap1 and Nrf2 were quantified by qPCR, which was conducted on Mastercycler^®^ ep realplex (Eppendorf, Germany). Each PCR system contained cDNA (15 ng) and 500 nM of specific primers for the target mRNA, and the reaction was catalyzed by Power SYBR Green PCR Master Mix, according to the manufacturer’s instruction. Cycling conditions were as follows: 50 °C for 2 min, 95 °C for 10 min, 40 cycles of 95 °C for 15 s, and 60 °C for 1 min. Keap1 and Nrf2 expression was detected with the amplification by 40 cycles. The following primers were used: 5′-GACAGCCTCTGACAACACAAC-3′ (forward for Keap1), 5′-GAAATCAAAGAACCTGTGGC-3′ (reverse for Keap1); 5′-CCTCAACTATAGCGATGCTGAATCT-3′ (forward for Nrf2), 5′-AGGAGTTGGGCATGAGTGAGTAG-3′ (reverse for Nrf2); 5′-AGAGCTACGAGCTGCCTGAC-3′ (forward for β-actin), 5′-AGCACTGTGTTGGCGTACAG-3′ (reverse for β-actin). Data were analyzed by GraphPad Prism software with β-actin as a reference gene, and its expression was not altered by any of the treatments. The relative expression levels of each target were calculated based on the cycle threshold (Ct) method, as described by Voelker et al. [32].

### 2.7. Statistical Analysis

The values obtained from the following tests, MTT assay, apoptosis, cell cycle and qRT-PCR, were analyzed by analysis of variance (One-way ANOVA, Dunnett’s multiple comparisons test) using GraphPad Prism 6.0 (GraphPad Software Inc. San Diego, CA, USA). The level of statistical significance was defined as *p* ≤ 0.05. To describe the type of pharmacokinetic interactions between extracts and D, the combination index (CI) was calculated, and data from MTT assay were employed. The values of CI being lower, equal, or higher than 1 (CI < 1, CI = 1, CI > 1) indicated the synergistic, additive, and antagonistic effect, respectively.

## 3. Results

### 3.1. Identification of Compounds in the Extracts

The results of spectrophotometric measurement of total phenolics and flavonoids content of the extracts were expressed as equivalents of gallic acid per g of dry extract (eq GA/g DE) and equivalents of quercetin per g of dry extract (eq Querc/g DE), respectively. They were determined to be 282.8 ± 73 mg eq GA/g DE and 306.9 ± 43 mg eq GA/1g DE, and 28.9 ± 0.5 mg eq Querc/1g DE and 38.5 ± 2.0 mg eq Querc/1g DE, for POA and PEA, respectively. The comparison of data concerning phenolics content of POA and PEA is presented in Table 1. The results of the LC-MS/MS analysis (Figure 1) showed that both extracts are rich in phenolic acids and flavonoids. POA is rich in quinic acid (8.72 mg/g DE), kaempherol-3-*O*-glucoside (1.33 mg/g DE), quercetin-3-*O*-glucoside (1.38 mg/g DE), and quercetin-3-*O*-galactoside (3.02 mg/g DE). PEA is characterized by a high content of aglycone, such as quercetin (5.50 mg/g DE) and a high content of quercetin derivatives: quercetin-3-*O*-galactoside (11.90 mg/g DE), quercetin-3-*O*-L-rhamnoside (9.79 mg/g DE), and quercetin-3-*O*-glucoside (1.49 mg/g DE). PEA is also rich in free gallic acid (3.49 mg/g DE) and epigallocatechin gallate (1.28 mg/g DE).

### 3.2. Cytotoxicity and Drug Synergism Analysis of Herbal Extracts and Doxorubicin

The evaluation of the cytotoxic effect of herbal extracts, alone and combined with D, was conducted on HepG2 cells. The IC_25_ and IC_50_ values of extracts and D, determined from the dose–response curves (Figure 2A–C), are presented in Table 2. Applied individually, PEA was more effective against HepG2 cells than POA. Applied in a mixture, in lower tested concentrations, POAD induced higher sensitivity of HepG2 cells than PEAD. To quantify the mode of interaction between tested substances, the combination index (CI) was calculated for IC_25_ and IC_50_ concentrations. Remarkable synergism for both mixtures, POAD (CI =0.62 and 0.13) and PEAD (CI = 0.89 and 0.39), was detected. Thus, the concentration required to inhibit cell viability for 25% and 50% for both agents in the mixtures has been remarkably reduced.

### 3.3. Effect of Herbal Extracts and Doxorubicin on Apoptosis and Cell Cycle

To determine whether the cytotoxicity of individual agents and their combinations is related to apoptosis and mediated by cell cycle arrest, the flow cytometry was applied. A significant increase in early apoptosis was determined after treatment with POA (21%). (Figure 3A,B). In addition, both concentrations of D, used for the preparation of combinations with POA and PEA, also increased early apoptosis of cancer cells (36% and 48.65%). An increase in both early and late apoptosis was observed when the PEA (29% and 38%, respectively) and POAD (26% and 17% respectively) were applied. However, in the case of PEAD co-treatment, only an increase in late apoptosis was detected (46.51%).

The analysis of the cell cycle phase distribution of HepG2 treated cells showed a significant cell cycle arrest in G2/M phase when individual treatment, POA (33%), and both concentration of D (41% and 43.45%), as well as co-treatments POAD (46%) and PEAD (42.69%) were applied (Figure 3C,D). Additionally, a significant increase in HepG2 cells in S phase was observed after individual treatment with PEA (37.4%) and co-treatment with PEAD (31.84%).

### 3.4. Effect of Herbal Extracts and Doxorubicin on Keap1 and Nrf2 Genes Expression

In malignant cells, alterations of the expression of Keap1 and Nrf2 genes are not rare. Here, the expression of Nrf2 and Keap1 was examined in HepG2 cells. Both co-treatments significantly increased Keap1 and simultaneously decreased Nrf2 gene expression (Figure 4).

## 4. Discussion

Polygonaceae species are rich sources of valuable secondary metabolites, mainly flavonoids. Data concerning chemical composition of *P. aviculare* is abundant, while *P. amphibia* has been less studied. Several studies gave important contributions in elucidating chemical composition of *P. aviculare* extracts [1,33,34,35,36,37,38,39,40,41,42], but Granica et al. [43,44,45], and Cai et al. [46] stands out. Granica et al. [43,44] focused on developing new standardization HPLC methods for *P. aviculare*. Taking into account the results of Granica [43], we have focused on a targeted search for given flavonol (myricetin (M), quercetin (Q), kaempferol (K), isorhamnetin (IR), kaempferide (KD)) glucuronides (U) and their acetylated derivatives (acU), as some of these tend to be the major compounds occurring in *P. aviculare* (Q-3-*O*-glucuronide or kaempherol-3-*O*-glucuronide). For HPLC separation conditions used in our work there is a pattern of elucidation order, as follows: MU, QU, MacU, KU, IRU, QacU1, QacU2, KacU1, IRacU1, KacU2, IRacU2, KDU, KDacU (Figure 5 and Figure 6) This way of analysis revealed significant differences between *P*. *aviculare* and *P*. *amphibia* plant extracts. Although both plant extracts contain myricetin-glucuronide and quercetin-glucuronide in significant amounts (not quantified), in PEA extracts the following compounds were not found: MacU, IRU, QacU1, QacU2, KacU1, IRacU1, KacU2, IRacU2, KDU, KDacU. Considering the scarce data on chemical composition of *P. amphibia* [34,35,42,47] this work, with quantitative and tentative qualitative HPLC analysis, creates a notable contribution.

On the other hand, to overcome the problem of overall toxicity and resistance of cancer cells to chemotherapeutics, a combined approach that employs both commercial cytostatic and herbal extracts was subjected to the analysis. The benefit of this approach involves the induction of more diverse mechanisms of action that hinder the development of resistance and allow for the reduction of cytostatic doses. In this study, we examined in vitro cytotoxic properties of POA and PEA, alone and combined with commercial cytostatic D. Our investigation provided corroborative evidence that POA and PEA extract could potentiate D cytotoxicity in hepatocarcinoma (HepG2) cells. This result is in accordance with our previous findings [28], which demonstrated a synergistic interaction between *Polygonum maritimum* extract and D in HepG2 cells. Likewise, Ghazali et al. [48] demonstrated that herb extracts obtained from *Polygonum minus* have an antiproliferative effect on HepG2 cells. It has also been reported that the *Polygonum cuspidatum* extract has an antiproliferative effect on hepatocarcinoma cells Bel-7402 and Hepa 1–6 [49], whereas extracts obtained from *Polygonum glabrum* and *Polygonum orientale* exhibited the protective activity on normal hepatocytes in vivo [50,51].

It is well known that inhibition of cancer cell proliferation may be a result of a pro-apoptotic effect and a cell cycle disruption. Consequently, the effect of extracts and their combinations with D on apoptosis and cell cycle arrest were monitored. Herein, it was confirmed that apoptosis induction plays an important role in D-induced cell death of HepG2 cells. Also, all treatments, but particularly with POA, applied alone and in a combination, were capable of inducing early or late apoptosis. Similar to this finding, Habibi et al. [7] have reported that methanol extract of *P. aviculare* induced apoptosis in breast cancer MCF-7 cells. As for the impact of *Polygonum* spp. extracts on the molecular mechanism of apoptosis, up-regulation of the apoptotic gene *p53* and down-regulation of the anti-apoptotic *Bcl-2* gene were demonstrated [7]. The pro-apoptotic effect controlled by p53 is accompanied by cell cycle arrest in G2/M phase [52]. Further on, the arrest of the cell cycle in G2/M phase is a well-established feature of D [53]. Importantly, the G2/M checkpoint serves to prevent damaged cells from entering mitosis and proliferate. Not only D, but also the extracts, particularly POA alone and combined with D, increased the number of HepG2 cells in the G2/M phase. Likewise, recent studies have shown that various *Polygonum* spp. extracts and their active compounds induced the pro-apoptotic effect and arrested HepG2 cells in G2/M phase [54,55]. Moreover, this study demonstrated that PEA alone and combined with D induced cell cycle arrest in the S phase. The obtained results are in line with Ghazali et al. [48], implying that *Polygonum minus* extracts induced S phase cell cycle arrest in HepG2 cells. Therefore, the observed synergism between D and tested extracts in cancer cells might be attributed to the interference with pathways involved in the regulation of apoptosis and cell cycles.

Data reported in the literature indicates that pro-apoptotic effects and cell cycle arrest induced by plant extracts could be attributed to their chemical composition. Searching for possible active compounds among the main constituents of the tested extracts pointed to free gallic acid, as well as quercetin and its derivatives, since they have been well documented to possess cytotoxicity linked to pro-apoptotic effects and the ability to induce cell cycle arrest [56,57,58]. Thus, quercetin caused the cell cycle arrest in G2/M phase, which was followed by a decrease in cell numbers in the G0/G1 phase [58]. Furthermore, gallic acid induced cell cycle arrest in malignant cells, contributing to inhibition of cancerous cell proliferation [59]. Moreover, quercetin induced apoptosis in various cancer cells [60,61].

Beside the growth-inhibiting and apoptosis-inducing effects, phytochemicals are capable of modulating Nrf2 expression. Furthermore, due to the overexpression of Nrf2, malignant cells are frequently highly resistant to different chemotherapeutics [19,21]. Therefore, Nrf2 is an important pharmacological target of effective chemotherapy. This study demonstrated that both co-treatments decreased Nrf2 expression in HepG2 cells. As expected, this response was followed by increased Keap1 gene expression. Similar results were obtained for resveratrol, an active compound of *Polygonum cuspidatum*, which was shown to modulate Nrf2 expression in a concentration- and time-dependent way [62]. Comprehensively observed, Nrf2 has a functional link with numerous genes reported to play specific roles in the development of drug resistance. For instance, Nrf2 influences the regulation of phase II-detoxifying enzymes, antioxidant defense enzymes, and multidrug resistance-associated proteins 1-6 (MRP 1-6) [23,63]. Altogether, regulation of Nrf2 is responsible, at least partially, for chemotherapy resistance, indicating the importance of determining Nrf2 inhibitors, such as POAD and PEAD.

In conclusion, since synergistic cytotoxicity in hepatocarcinoma cells was observed, the combined approach that employs *Polygonum aviculare* and *Persicaria amphibia* ethanol extracts and cytostatic D, could serve as a good starting point in the search for hepatocarcinoma chemotherapy improvement.

## Figures and Tables

**Figure 1 foods-10-00811-f001:**
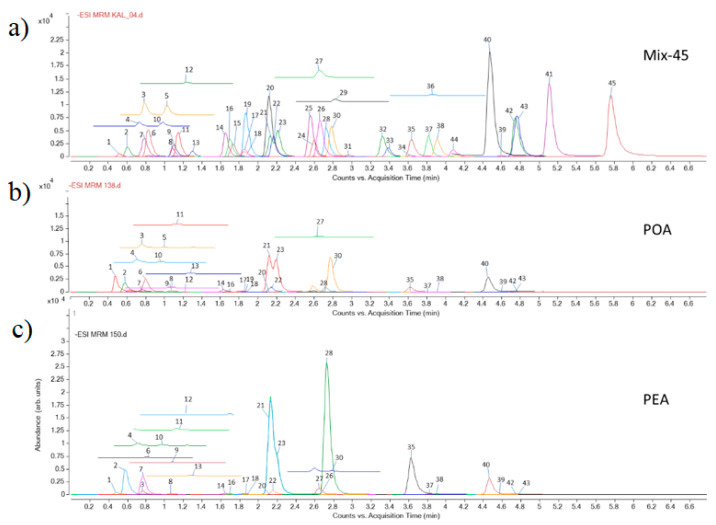
MRM chromatograms of standard compounds (3.125 μg/mL each standard), (**a**); of *P. aviculare* herb ethanol extract, POA, (**b**); and of *P. amphibia* herb ethanol extract, PEA, (**c**); 1: Quinic acid, 2: Gallic acid, 3: Protocatechuic acid, 4: Catechin, 5: 2,5-dihydroxybenzoic acid, 6: 5-*O*-caffeoylquinic acid, 7: Epigallocatechin gallate, 8: *p*-Hydroxybenzoic acid, 9: Esculetin, 10: Epicatechin, 11: Caffeic acid, 12: Vanillic acid, 13: Syringic acid, 14: *p*-Coumaric acid, 15: Umbelliferon, 16: Scopoletin, 17: Ferulic acid, 18: Sinapic acid, 19: Vitexin, 20: Luteolin-7-*O*-glucoside, 21: Quercetin-3-*O*-galactoside, 22: Rutin, 23: Quercetin-3-*O*-glucoside, 24: *o*-Coumaric acid, 25: Apiin, 26: Apigenin-7-*O*-glucoside, 27: Myricetin, 28: Quercetin-3-*O*-L-rhamnoside, 29: Secoisolariciresinol, 30: Kaempferol-3-*O*-glucoside, 31: 3,4-dimethoxycinnamic acid, 32: Baicalin, 33: Daidzein, 34: Matairesinol, 35: Quercetin, 36: Cinnamic acid, 37: Naringenin, 38: Luteolin, 39: Apigenin, 40: Kaempferol, 41: Baicalein, 42: Isorhamnetin, 43: Chrysoeriol, 44: Genistein, 45: Amentoflavone.

**Figure 2 foods-10-00811-f002:**
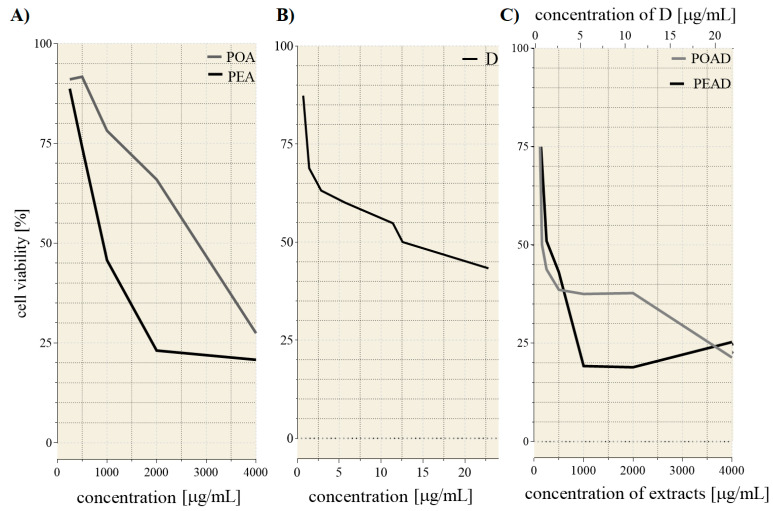
Inhibition rates of HepG2 cells treated with individual extracts (**A**); doxorubicin (**B**); and their combination (**C**) after 24 h.

**Figure 3 foods-10-00811-f003:**
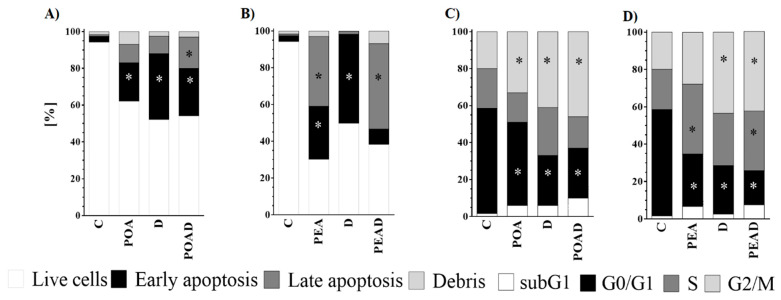
Analysis of apoptosis (**A**,**B**) and cell cycle arrest (**C**,**D**) in HepG2 cells after treatment with extracts, doxorubicin (**D**), and their combinations. An increase in the number of early and late apoptotic cells is always at the expense of living cells; the results are expressed as percentages compared to the untreated control * *p* ≤ 0.05.

**Figure 4 foods-10-00811-f004:**
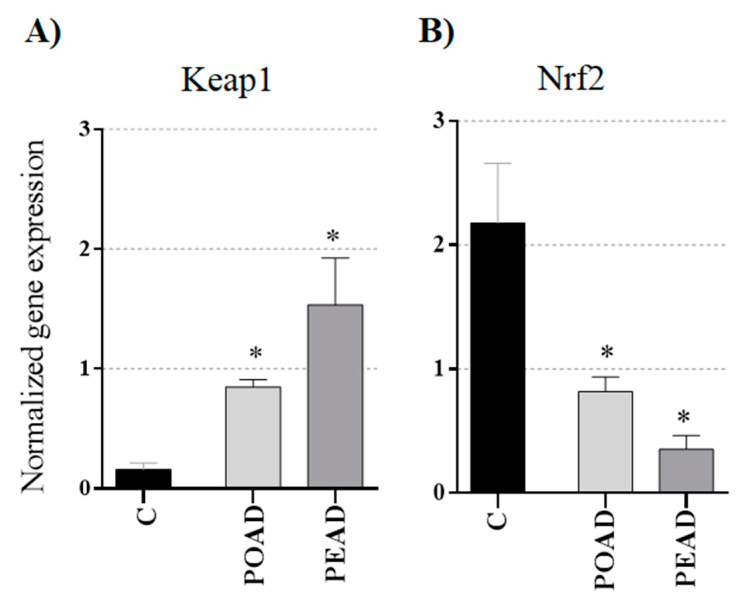
The effect of extracts and doxorubicin (D) combined (IC_25_) on the expression of (**A**) Keap1 and (**B**) Nrf2 genes in HepG2 cells evaluated by the qRT-PCR, * *p* ≤ 0.05.

**Figure 5 foods-10-00811-f005:**
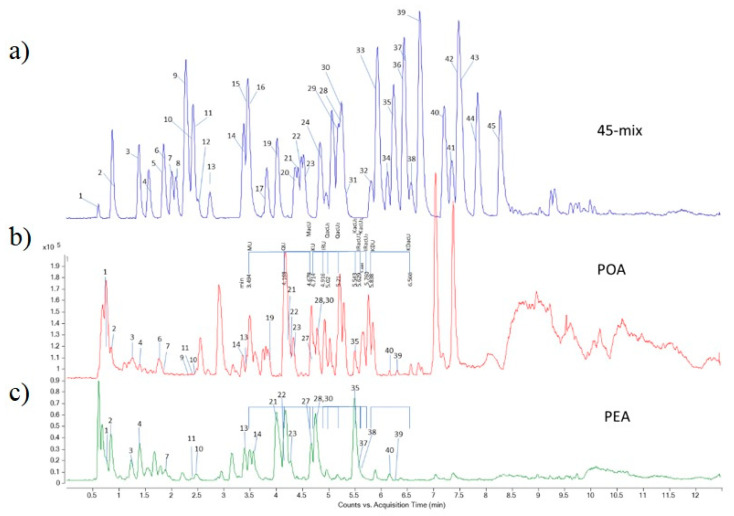
ESI BPC chromatograms, negative ionization mode (MS2Scan) of standard mix-45 (1.56 μg/mL, each compound), (**a**); of *Polygonum aviculare* ethanol herb extract, POA, (**b**); and *Persicaria amphibia* ethanol herb extract, PEA, (**c**); with labeled phenolics that were confirmed by quantitative LC-MS/MS analysis, and tentatively determined -glucuronides (U), and acetylglucuronides (acU) derivatives of Myricetin (M), Quercetin (Q), Kaempferol (K), Isorhamenetin (IR) and Kaempferide (KD), e.g., MU-myricetin-glucuronide, MacU-myricetinacetylglucuronide (Appendix A).

**Figure 6 foods-10-00811-f006:**
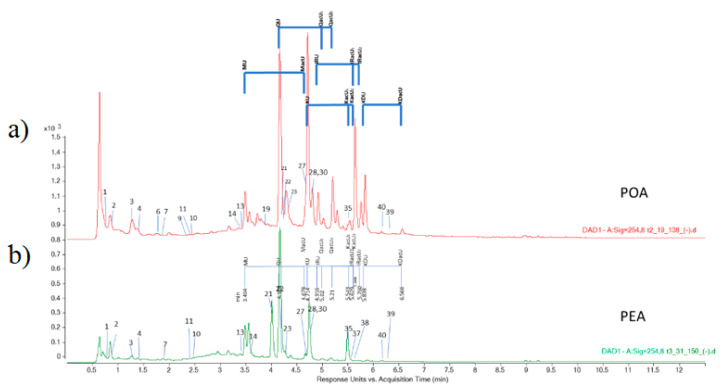
HPLC-DAD chromatograms (255 nm) of *Polygonum aviculare* ethanol herb extract, POA, (**a**); and *Persicaria amphibia* ethanol herb extract, PEA, (**b**); with labeled phenolics that were confirmed by quantitative LC-MS-MS analysis. Tentatively determined -glucuronides (U), and acetylglucuronides (acU) derivatives of Myricetin (M), Quercetin (Q), Kaempferol (K), Isorhamenetin (IR) and Kaempferide (KD) are also labeled, e.g., MU-myricetin-glucuronide, MacU-myricetinacetylglucuronide (Appendix A).

**Table 1 foods-10-00811-t001:** Concentrations of phenolics found in *Polygonum aviculare* (POA) and *Persicaria amphibia* (PEA) ethanol extracts (expressed as μg of phenolics per gram of dry extract).

Class of Secondary Metabolites	Compound	No ^a^	Rt ^b^ [min]	LoQ ^c^ [μg/g de]	Content [μg/g dw] ^d^	
					POA	PEA
Cyclohexanecarboxylic acids	Quinic acid	1	0.52	5.0	(8.7 ± 0.9) 10^3^	(8.8 ± 0.9) 10^2^
Hydroxybenzoic acids	Gallic acid	2	0.58	10	(9.5 ± 0.8) 10^2^	(3.5 ± 0.3) 10^3^
	Protocatechuic acid	3	0.79	2.0	(2.3 ± 0.2) 10^2^	(1.9 ± 0.1) 10^1^
	2,5-dihydroxybenzoic acid	5	1.03	3.5	(3.0 ± 0.2) 10^1^	<LoQ
	*p*-Hydroxybenzoic acid	8	1.08	4.0	(3.1 ± 0.2) 10^1^	(4.8 ± 0.3) 10^1^
	Vanillic acid	12	1.24	50	(2.6 ± 0.8) 10^1^	(0.3 ± 0.1) 10^2^
	Syringic acid	13	1.31	20	(1.5 ± 0.3) 10^2^	(1.1 ± 0.2) 10^2^
Phenylpropanoids	Cinnamic acid	36	3.91	40	<LoQ	<LoQ
Hydroxycinnamic acids	Caffeic acid	11	1.18	3.0	(2.2 ± 0.2) 10^1^	(6.2 ± 0.4) 10^1^
	*p*-Coumaric acid	14	1.69	2.0	(5.1 ± 0.4) 10^1^	(3.7 ± 0.3) 10^1^
	Ferulic acid	17	1.90	5.0	(2.8 ± 0.3) 10^1^	(5.0 ± 0.5) 10^1^
	Sinapic acid	18	1.92	20	(1.6 ± 0.2) 10^1^	(3.9 ± 0.4) 10^1^
	*o*-coumaric acid	24	2.62	3.0	1.3 ± 0.1	<LoQ
	3,4-dimethoxycinnamic acid	31	2.99	25	<LoQ	<LoQ
Chlorogenic acids	5-*O*-caffeoylquinic acid	6	0.80	3.5	(6.9 ± 0.3) 10^2^	11.0 ± 0.5
Flavan-3-ols	Catechin	4	0.74	25	(7.0 ± 0.7) 10^2^	(6.5 ± 0.6) 10^2^
	Epicatechin	10	0.95	30	(8.8 ± 0.9) 10^1^	(2.3 ± 0.2) 10^2^
Flavan-3-ol-derivatives	Epigallocatechin gallate	7	0.81	50	(1.5 ± 0.1) 10^2^	(1.3 ± 0.1) 10^3^
Coumarins	Esculetin	9	1.13	3.0	5.1 ± 0.3	9.1 ± 0.5
	Umbelliferone	15	1.73	5.0	<LoQ	<LoQ
	Scopoletin	16	1.77	3.5	1.0 ± 0.1	11.2 ± 0.9
Flavone glycosides	Luteolin-7-*O*-glucoside	20	2.13	2.5	(11.0 ± 0.3) 10^−1^	3.1 ± 0.1
	Vitexin	19	1.90	2.0	3.2 ± 0.2	<LoQ
	Apiin	25	2.60	1.5	<LoQ	<LoQ
	Apigenin-7-*O*-glucoside	26	2.81	3.0	<LoQ	(6.0 ± 0.3) 10^−1^
	Baicalin	32	3.4	10	<LoQ	<LoQ
Flavones	Luteolin	38	4.03	2.0	6.4 ± 0.3	12.6 ± 0.6
	Apigenin	39	4.71	5.0	(8.5 ± 0.6) 10^1^	(4.0 ± 0.3) 10^1^
	Baicalein	41	5.15	15	<LoQ	<LoQ
	Chrysoeriol	43	4.82	2.0	(5.0 ± 0.1) 10^−1^	(8.0 ± 0.4) 10^−1^
Biflavonoid	Amentoflavone	45	5.78	2.5	<LoQ	<LoQ
Flavonol-glycosides	Quercetin-3-*O*-galactoside	21	2.16	3.0	(3.0 ± 0.2) 10^3^	(11.9 ± 0.7) 10^3^
	Quercetin-3-*O*-rutinoside	22	2.33	1.5	(26.2 ± 0.8) 10^1^	(20.5 ± 0.6) 10^1^
	Quercetin-3-*O*-glucoside	23	2.25	2.0	(13.8 ± 0.4) 10^2^	(14.9 ± 0.4) 10^2^
	Quercetin-3-*O*-L-rhamnoside	28	2.75	1.5	(1.6 ± 0.1) 10^2^	(9.8 ± 0.6) 10^3^
	Kaempferol-3-*O*-glucoside	30	2.8	2.0	(13.3 ± 0.5) 10^2^	(2.8 ± 0.1) 10^1^
Flavonols	Myricetin	27	2.67	50	(11.1 ± 0.8) 10^1^	(8.6 ± 0.6) 10^2^
	Quercetin	35	3.74	50	(0.4 ± 0.1) 10^3^	(5.5 ± 1.6) 10^3^
	Kaempferol	40	4.55	3.0	(12.0 ± 0.8) 10^1^	(13.4 ± 0.9) 10^1^
	Isorhamnetin	42	4.79	10	5.1 ± 0.3	8.6 ± 0.5
Flavanones	Naringenin	37	3.87	3.5	6.5 ± 0.4	(1.6 ± 0.1) 10^1^
Isoflavones	Daidzein	33	3.43	5.0	<LoQ	<LoQ
	Genistein	44	4.12	3.0	<LoQ	<LoQ
Lignans	Secoisolariciresinol	29	2.90	25	<LoQ	<LoQ
	Matairesinol	34	3.66	50	<LoQ	<LoQ
TOTAL					18,782 (1.88%)	37,113 (3.71%)

^a ^Numbers are used as labels on given chromatograms bellow. ^b^ From the method validation published in Orčić et al. [27]. ^c^ Calculated from the instrument quantification limit (Orčić et al. [27]) and sample dilution. ^d^ Results are given as the concentration (μg/g of dry extract) ± relative standard deviation of repeatability (as determined by method validation [27]). LoQ–limit of quantitation; the standard curves were provided in Supporting materials (Appendix A).

**Table 2 foods-10-00811-t002:** The cytotoxicity of herbal extracts and doxorubicin (D), either alone or in a two-drug combination on HepG2 cells.

Individual Treatments			
	IC_25_ *	IC_50_*	
D	1.3	12.56	
POA	1250	2800	
PEA	500	910	
IC_25_ * values of the co-treatments			
POAD		PEAD	
POA	D	PEA	D
120	0.68	140	0.79
IC_50_ * values of the co-treatments			
**POAD**		**PEAD**	
POA	D	PEA	D
**160**	**0.91**	**250**	**1.43**
CI			
	POAD	PEAD	
IC_25_	0.62	0.89	
IC_50_	0.13	0.39	

* The concentrations are expressed in µg/mL. *Polygonum aviculare* ethanol extract (POA); *Persicaria amphibia* ethanol extract (PEA); Doxorubicin (D); Co-treatment of POA and D (POAD); Co-treatment of PEA and D (PEAD); Combination index (CI). The concentrations in bold, individually and combined, were used in flow cytometry analysis.

## Data Availability

The data presented in this study are available on request from the corresponding author.

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
