# Peer review of "In Vitro Study of Two Edible Polygonoideae Plants: Phenolic Profile, Cytotoxicity, and Modulation of Keap1-Nrf2 Gene Expression"

_foods, 2021, doi:10.3390/foods10040811_

Round 1

Reviewer 1 Report

The manuscript entitled “In vitro study of two edible Polygonoideae plants: Chemical characterization, cytotoxicity, and modulation of Keap1-Nrf2 gene expression” regards chemical characterization and hepatorprotective effects of P. aviculare and P. amphibia ethanolic extract against hepatocelular carcinoma cells.

The article was well written. Although the paper has relevant information about the beneficial effect of both extracts, the analytical procedures of the phytochemical quantitation should be completed. For this reason, I think that additional information should be included to improve the results displayed.

Introduction

Line 64 and 67: Please standarize the way to express Nrf2, since, it appears in italics in some sentences and  without italics in other ones.

Line 74: Keap1 or Keap1. Please standarize their expression.

Material and Methods

Line 80: The authors should specify the phenolic compounds standards that they used in analytical procedures.

Line 102: Authors should specify in more detail how they obtained the extract, for example, extraction time, stirring or temperature. Was the obtained extract filtered after extraction process and prior vacuum driying?

Line 104: Which phytochemical o phytochemical group wanted to purify the liquid-liquid extraction process?

Line 114: The authors should explain in more detail the condition of the HPLC procedure (elution gradient, mobile phases, temperatura…) as well as the source parameters and detection of MS/MS assay. Moreover, author should describe the method to quantify the phytochemical compounds.

Results

Line 202: Did the author perform Follin Ciocalteau to determine the total phenolic content? It is not clear the way to quantify the total phenolic content, because in the material and method section was not specified.

Table 1: In my opinion, the standar deviation should be standarized. In this sense, the significant numbers have to be adjusted to the first significant number unequal  to zero. With the actual expression, it is hard to understand the quantitation reproducibility.

Did the authors develop any validation of the quantitation method applied? I think that it is necessary to understand the limit of detection and quantification of the analytical method applied.

Line 208: It was not detailed if the authors applied the same standards to quantify the compounds contained in the extract or if they performed a quantifitation applying some of them.

Reviewer 2 Report

The manuscript deals with an interesting subject.

In this study, were exploted the cytotoxic properties of ethanol extracts of aerial parts of P. aviculare and P. amphibia. A synergistic cytotoxicity in hepatocarcinoma cells was observed. Also, ethanolic extracts were characterized.

I suggest major revision

Comments

  1. A more detailed presentation of the chemical analysis results could enhance the current research in the style of this journal. I recommend you to display two characteristic chromatograms of each plant extract, which show the main compounds. Also, include the retention times, the theoretical and the experimental ion m/z and mass error (ppm). Finally, calibration curves can be plotted too. Could you suggest a way an application of these extracts to food?
  2. The authors are referred to chemical characterization of two edible Polygonoideae plants, however in the manuscript they have quantified by LC-MS only flavonoids and phenolic acids and they haven’t report other chemical substances that may be present in the extracts. Therefore, the term “chemical characterization” is not representable and should be replaced to “characterization of phenolics”.
  3. Lines 102-106: Have the authors optimize the extraction of procedure? Did they perform the extraction in triplicates? Why did they use 80% ethanol for the extraction of flavonoids and phenolic acids? Did the authors follow or modified any specific literature protocol? If so, relevant references must be mentioned.
  4. Lines 116-121: As above, please report if you have followed or modified any specific literature protocol and provide references.
  5. Subsection 3.1: In Table 1 the authors use “Nd” for the concentration of compounds lower than the detection limit. Please report either in Subsection 3.1 or in Supporting Information, the detection and quantification limits, r2, retention times and representative chromatogram for both plants. Regarding quantification results and the relative standard deviation, the number of significant figures is indicative of the repeatability of the measurement, please report how many times did you performed the analysis and correct accordingly (triplicates?). The characterization of total and individual phenolics in Polygonum aviculare and Persicaria amphibia has been performed by other researchers. I recommend more comparison with literature.
  6. Line 329-In conclusion…: Regarding pro-apoptotic effects and cell cycle arrest, taking into consideration that PEA extract has higher concentrations of free gallic acid, quercetin and its derivates from POA, which present cytotoxicity linked to pro-apoptotic effect (lines 306-314), can we conclude that PEA presented better results against POA?

Round 2

Reviewer 1 Report

The revised version of manuscript entitled “In vitro study of two edible Polygonoideae plants:Phenolic profile, cytotoxicity and modulation of Keap1-Nrf2 gene expression” was extensively improved since the lack of information was corrected and supported with necessary information by authors.

However, minor modifications are still necessary.

Line 164: If author would use ESI to mention electrospray ionization, they should specify it before (line 138).

Line 144: Please correct N2.

Table 1: In spite of include additional information about LoQ and a more detailed description of phytochemical composition, the standard deviation is still not uniformed expressed. Moreover, it would be interesting to add a new column that specifies the presence of the phytochemicals in each of the matrices studied.

Reviewer 2 Report

The authors responded in some comments and suggestions, but not in all.

Information about the theoretical and the experimental ion m/z and mass error (ppm) were not given. It is also not yet clear whether the extractions and analyzes were performed in triplicates.

The authors in line 69, insist referring to chemical characterization. It’s difficult for me to find the correlation between this study and the scope of this journal.
